# Experimental and Numerical Studies on Preparation of Thin AZ31B/AA5052 Composite Plates Using Improved Explosive Welding Technique

**Qi Wang [1], Xuejiao Li [1,2,3,*], Biming Shi [1] and Yong Wu [2]**

[1]   College of Mining and Safety Engineering, Anhui University of Science & Technology,
      Huainan 232001, China; qwang162@126.com (Q.W.); pytraetq@126.com (B.S.)

[2]   College of Chemical Engineering, Anhui University of Science & Technology, Huainan 232001, China;
      wuyong15705543189@163.com

[3]   Anhui Leiminghkehua Limited Liability Company, Huaibei 235000, China

[*]   Correspondence: xjli@aust.edu.cn; Tel.: +86-152-5698-5499

**Abstract:** In this work, an improved explosive welding technique was investigated to fabricate a thin Mg/Al plate, where an additional thin aluminum sheet was used as a buffer layer between the explosive and the Al plate, and the Mg plate was rigidly constrained by a steel plate to avoid excessive deformation. Moreover, the welding parameters were optimized using theoretical analysis and numerical simulation, and the interfacial behavior was simulated using the SPH method. The bonding properties of the achieved joints were investigated using microstructure observation and mechanical tests. It was concluded that this technique is an effective method for producing a thin Mg/Al composite plate. In both morphology observation and mechanical tests, an excellent bonding quality was confirmed. In addition, smoothed particle hydrodynamics (SPH) simulation revealed an extreme condition of local high temperature and plastic strain in the welding process, and the characteristic parameters of waves obtained using simulation are well congruous with the experiment.

**Keywords:** explosive welding; magnesium; aluminum; thin plate; numerical simulation

---

## 1. Introduction

Magnesium alloy is one of the lightest structural metal materials, showing the advantages of high specific strength, specific stiffness, good damping property and electromagnetic shielding performance [1,2]. Thus, magnesium alloys are promising materials in industries such as automobile, aerospace and electronics and also play an important role in developing a new type of environment-friendly metal material [3]. However, the poor corrosion and wear resistance limits its further application. In contrast, aluminum alloys exhibit excellent corrosion resistance and good mechanical properties [4,5]. Therefore, the layered composites of magnesium and aluminum alloys are promising structural materials, due to combining the advantages of the two metallic materials and giving the opportunity to provide the unique functional and operational properties of the layered materials [6]. However, obtaining a high quality Mg/Al joint is still a challenging task, due to potential technical problems such as poor solubility of Mg in Al and formation of intermetallic compounds [7,8].

Up to now, several methods such as diffusion bonding [9,10], friction stir welding [11], hot roll bonding [12–14], and laser welding [15] have been attempted to join aluminum and magnesium. However, due to the strong heat input in these technologies, brittle intermetallic compounds are easy to generate at the Mg/Al bonding interface, which greatly degrade the mechanical properties of the bonding interface. Unlike these technologies, explosive welding is characterized by a solid state method for joining various layered metal materials, which produces a metallurgical bond by

high-speed oblique collision between two layers of metal, with the help of energy of explosion [16,17]. Because of the instantaneity of the explosion, the characteristic time of the explosive welding process was evaluated to be in the order of $10^{-6}$ s, which can effectively prevent large-scale melting at the bonding interface by limiting heat transfer [18]. Thus, explosive welding is able to keep the initial material properties of the parent materials invariable and most importantly minimize the formation of intermetallic compounds. In addition, explosive welding is characterized by formation of high-speed jets before the bonding process, which can remove the surface oxide film that has a negative effect to the formation of the metallurgical bond [19]. Therefore, explosive welding is extremely suitable for joining such Mg/Al dissimilar bulk metals.

Due to the high bonding ability, explosive welding has drawn increasing interest in the manufacturing of aluminum and magnesium. Zeng et al. [7] investigated the effect of inert gas-shielding on the interface and mechanical properties of Mg/Al composite plate and found that explosive welding under inert gas atmosphere can effectively improve the weld quality. Zhang et al. [3] fabricated a AA6061/AZ31B composite cladding plate and carried out systematic research on the microstructure evolution and mechanical properties of the bonding interface. Fronczek et al. [20] prepared a three-layered A1050/AZ31/A1050 composite plate via explosive welding, where intermetallic phases of $Mg_2Al_3$ and $Mg_{17}Al_{12}$ were found in the local melting zone. Zhang et al. [21] studied the effect of annealing on the interface microstructure and mechanical characteristics of AZ31B/AA6061 explosive welding plates and found that intermetallic compounds of $Al_3Mg_2$ and $Al_{12}Mg_{17}$ tended to be generated at the bonding interface when the annealing temperature was at and above 250 °C. Recently, Inao et al. [22] explored a modified explosive welding technique to join a thin Al plate onto an AZ31 plate using a gelatin layer as the pressure-transmitting medium. However, all the reports referred to explosive welding of a thick Mg plate to a thick Al plate or a thick Mg plate to a thin Al plate. To our knowledge, there have been no studies concerning explosive welding of a thin Mg plate to a thin Al plate. In fact, thin Mg/Al composite plates have a broad industrial application prospect, and the combined method of explosive welding and multi-pass rolling must be introduced to obtain this product, which is characterized by extremely high cost [23]. The real reason for the absence of thin Mg/Al explosive welding composite plates is that it is difficult to produce thin composite plates with high surface quality using conventional explosive welding techniques, especially for magnesium alloy, which is characterized by poor ductility and wear resistance [23,24]. During the high-speed collision process of explosive welding, the thin Mg plate is easily broken or forms surface defects such as void and fold.

The aim of this work is to obtain a thin Mg/Al composite plate using explosive welding directly and give a better understanding on the welding process and interfacial behavior. To this end, numerical simulation (AUTODYN) accompanied by theoretical analysis was raised to optimize the parameters of explosive welding. Then, an improved explosive welding technique was carried out, where an additional thin aluminum sheet was used as a buffer layer between the explosive and the Al plate, and the Mg plate was rigidly constrained by a steel plate to avoid excessive deformation. After the welding test, the microstructures of the recovered samples were characterized using scanning electron microscopy (SEM), and the mechanical properties were measured using tensile and tensile-shear tests. Finally, the welding process was analyzed using the smoothed particle hydrodynamics (SPH) method.

## 2. Numerical Analysis

### 2.1. Numerical Arrangements for Explosive Welding

Collision velocity ($V_p$) and detonation velocity ($V_d$) are two significant parameters for explosive welding, an accurate control of them is considered as a guarantee for good welding quality, especially for thin plates welding. $V_d$ is recommended to be in a proper range, the minimum ($V_{dmin}$) was raised to ensure the formation of a wavy joint interface, which can be calculated through Equation (1) proposed by Cowan et al. [25]. $V_p$ is defined to promote metal plastic flow between plates, which can

be obtained using an empirical equation derived from Gurney's one-dimensional motion model, where the flyer plate was considered as a rigid body while its acceleration process was left out. Therefore, the calculated $V_p$ is actually the terminal velocity of the flyer plate. It is reliable enough for the prediction of thick plate welding, but not available for thin plates where bonding quality is sensitive to the $V_p$. Numerical simulation owns the advantage of predicting $V_p$ involving all the variables, which contributes to taking an accurate control of collision velocity.

$$V_{\text{dmin}} = \sqrt{\frac{2R_e(H_1 + H_2)}{\rho_1 + \rho_2}} \tag{1}$$

where $R_e$ is the critical Reynolds number, taking values of 10.6 [26], $H_1$ and $H_2$ are the Vickers hardness numbers of AA5052 and AZ31B, taking 650 MPa and 900 MPa, respectively, $\rho_1$ and $\rho_2$ equals 2.68 g·cm$^{-3}$ and 1.775 g·cm$^{-3}$, respectively.

In this work, AA5052 and AZ31B alloys were set as flyer plate and base plate, respectively, with the same dimension of 210 mm × 300 mm × 0.5 mm. Ammonium nitrate with fuel oil (ANFO) was used as the explosive material, with a detonation velocity of 2800 m·s$^{-1}$, slightly larger than the value 2705 m·s$^{-1}$ calculated by Equation (1). The flyer plate was protected by another Al foil with a thickness of 0.5 mm to avoid it being fragmented in the detonation process. To obtain a reasonable value of $V_p$ for the experiment, two-dimensional numerical simulation based on hydrodynamic AUTODYN-2D was executed. As shown in Figure 1, a planar model with element size of 0.1 × 0.1 mm was established for the analysis, where an ALE (Arbitrary Lagrange-Euler) processor was adopted for plates while a 2D multiple-material Euler processor was employed for air and the explosive. External gap was decided as the interaction algorithm. To describe their instantaneous state amid the detonation process, the ideal gas EOS (equation of state) was selected for the part of air, and the behavior of ANFO (ammonium nitrate fuel oil) explosive was modelled using the Jones-Wilkins-Lee (JWL) EOS relationship [27]:

$$P = A(1 - \frac{\omega\eta}{R_1})\exp(-\frac{R_1}{\eta}) + B(1 - \frac{\omega\eta}{R_2})\exp(-\frac{R_2}{\eta}) + \omega\eta\rho_0 e \tag{2}$$

where $\eta$ is $\rho/\rho_0$, and $A$, $B$, $R_1$, $R_2$ and $\omega$ are parameters of ANFO, taking 49.46 GPa, 1.89 GPa, 3.907, 1.118 and 0.33 [28], respectively.

Meanwhile, the Johnson-Cook equation was utilized as the constitutive law for Al and Mg alloys, and the Mie-Grüneisen type shock Hugoniot EOS was employed, which is expressed in Equation (3), taking strain-rate and strain-hardening effects into account.

$$P = P_H + \Gamma_0\rho_0(e - e_H) \tag{3}$$

$$P_H = \frac{\rho_0 c_0{}^2 \mu(1 + \mu)}{[1 - (s - 1)\mu]^2} \tag{4}$$

$$e_H = \frac{1}{2}\frac{P_H}{\rho_0}(\frac{\mu}{1 + \mu}) \tag{5}$$

where $P$ is the pressure, $\rho$ is the density, $\Gamma_0$ is the Grüneisen coefficient, taking 1.97 for Al and 1.43 for Mg; it is always assumed that $\Gamma_0\rho_0 = \Gamma\rho$. $e$ is the inner energy, $\mu = \rho/(\rho_0 - 1)$, $s$ is the material constant, taking 1.339 for Al and 1.26 for Mg, $c_0$ is the bulk sound velocity, with 5386 m·s$^{-1}$ for Al and 4516 m·s$^{-1}$ for Mg; they are constants of shock wave velocity (Us = $c_0$ + $s$Up) [29,30].

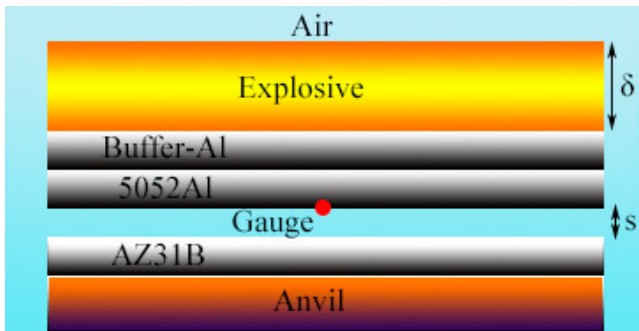

**Figure 1.** Diagram of numerical simulation arrangements.

The explosive thickness (δ) and stand-off distance (s) are two vital parameters to be determined for the explosive welding assembly and both have significant influence on $V_p$. Equation (6) is always utilized for the estimation of s [31], which correlates s and δ empirically.

$$s = 0.2(\delta + t) \tag{6}$$

where t is the thickness of the flyer plate, including buffer Al (0.5 mm) and 5052Al (0.5 mm) here.

To explore the optimum $V_p$ for the explosive welding of AA5052 and AZ31B alloy, four control groups (group 1–4) were set with explosive thickness δ 6, 6.5, 7 and 7.5 mm, respectively, based on the preliminary simulation results. The corresponding stand-off distance is 1.4, 1.5, 1.6 and 1.7 mm. To obtain the $V_p$ under these four conditions, a gauge was set below the flyer plate as shown in Figure 1, moving with its dynamic movement. Velocities of the gauge from group 1–4 were extracted and are depicted in Figure 2.

As shown in Figure 2, velocity-displacement curves well reflect the movement process of the flyer plate. With the initiating of explosives, the flyer plate was launched instantly to a speed of 450~520 m·s$^{-1}$, and then the speed value increases slowly with increasing displacement due to the detonation product action. $V_p$ was only obtained when the flyer plate struck the base plate, and the value of $V_p$ calculated under the four groups was 620, 681, 718 and 751 m·s$^{-1}$, respectively, which are positively correlated with δ and s.

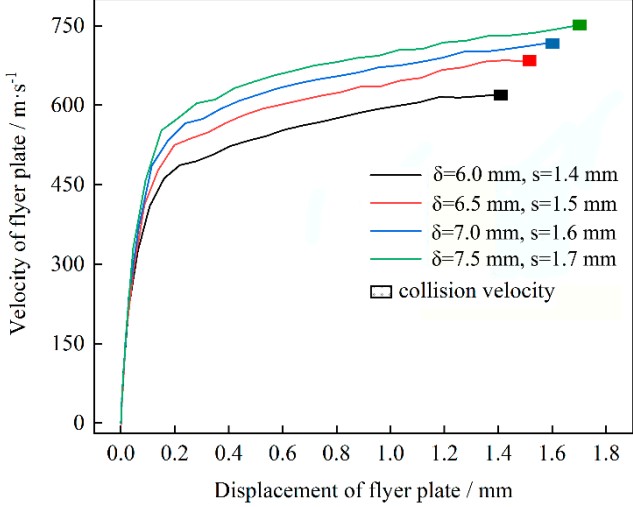

**Figure 2.** Velocity curves of gauge in the motion process.

## 2.2. Determination of Welding Parameters

To evaluate the feasibility of $V_p$ calculated from four setting groups, the explosive welding window was established to select the optimum welding parameters for the fabrication of AA5051/AZ31B bimetal foils (Figure 3). Collision angle $\beta$ and detonation velocity $V_c$ were set as coordinate axes of the window, respectively, and they are mutually connected with $V_p$ through Equation (7). The weldability window of these two alloys was developed according to previous research [32]. It is generally recognized that metallurgical bonding can be attained when the welding parameters are within the window. Though the four setting groups theoretically satisfy the demand, the parameters from group 1 are still recommended for the consideration of avoiding energy waste and eliminating the adverse effects caused by excessive heat input. The initial parameters decided are listed in Table 1.

$$2 \sin \frac{\beta}{2} = \frac{V_p}{V_c} \tag{7}$$

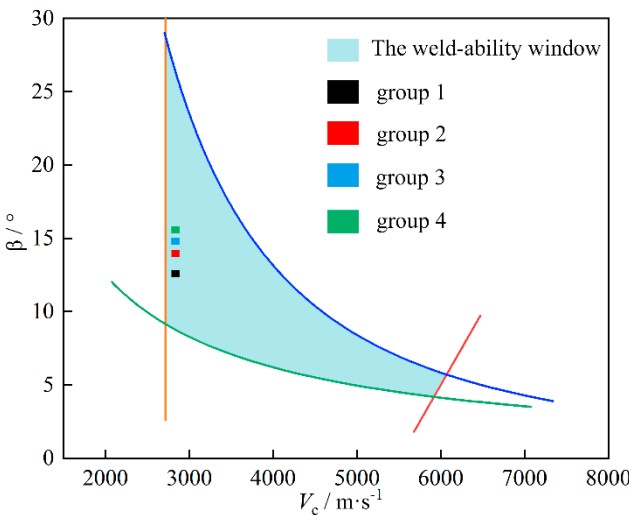

**Figure 3.** Weldability window of AA5051/AZ31B and collision velocities from setting groups.

**Table 1.** Initial parameters used for explosive welding.

| Explosive Thickness δ/mm | Stand-Off Distance s/mm | Collision Velocity $V_p$/m·s⁻¹ | Collision Angle β/° |
|---|---|---|---|
| 6.0 | 1.4 | 620 | 12.7 |

## 3. Experimental Procedure

### 3.1. Preparation of Materials

Experimental setting is in good agreement with that of group 1 in numerical simulation except for the explosive material; emulsion explosive was selected practically rather than the ANFO used in the simulation, because there is no proper constitutive law and EOS for emulsion explosive, while ANFO is a typical low-energy explosive that can be analogous to emulsion explosive in the same density and detonation velocity condition. The chemical compositions of AA5052 Al and AZ31B Mg alloy are given in Tables 2 and 3. Superficial roughness were both tested to be 0.2 μm (Ra) according to GB/T 1031-2009 [33]. Mg alloy was glued tightly with the anvil using epoxy resin. Surfaces cleaning were carried out using absolute ethyl alcohol before the welding process.

**Table 2.** Chemical composition of AA5052 aluminum alloy (wt.%).

| Element | Si | Cu | Mg | Zn | Mn | Cr | Fe | Al |
|---------|------|------|-----------|------|------|-----------|------|------|
| Content | ≤0.25 | ≤0.10 | 2.20–2.80 | ≤0.10 | ≤0.10 | 0.15–0.35 | ≤0.40 | Bal. |

**Table 3.** Chemical composition of AZ31B magnesium alloy (wt.%).

| Element | Al | Si | Mn | Ca | Zn | Mn | Fe | Cu | Mg |
|---------|-----------|------|------|------|---------|---------|-------|------|------|
| Content | 2.50–3.50 | 0.08 | 2.00 | 0.04 | 0.6–1.4 | 0.2–1.0 | 0.003 | 0.01 | Bal. |

### 3.2. Characterization Methods

After the explosive welding experiment, the macroscopic feature of the clad was checked using an ultrasonic fault detector (Reedea MIT-2300, Ningbo, China) with an image processing digital system; evaluation of the bonding morphology was processed later after that. Specimens from the bimetal plate were cut along the detonation direction, and they were ground up to 5000# with SiC paper before being polished to 0.25 μm using diamond paste. Scanning electron microscopy (SEM, Zeiss Gemini SEM 500, Jena, Germany) was employed for the investigation. To check the mechanical properties of the welded foil, tensile and tensile-shear tests were conducted in force control on a machine (MTS 809, Eden Prairie, MN, USA) at room temperature with a strain rate of $10^{-4}$/s according to GB/T6396-2008 [34]. The testing dimensions of the specimens are shown in Figure 4.

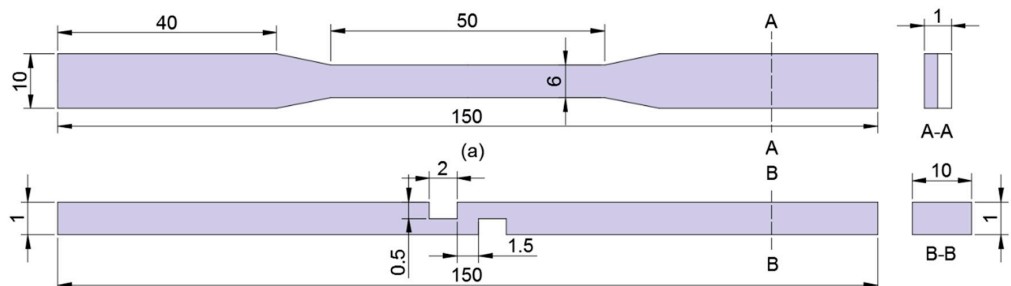

**Figure 4.** Dimensions of tensile and tensile-shear tests specimens.

## 4. Results and Discussion

### 4.1. Macroscopic Feature of AA5052/AZ31B Plate

Figure 5 shows the macro morphology of the recovered AA5052/AZ31B sample after explosion, with corresponding ultrasonic testing results. It is clear from Figure 5a that a good welding quality was obtained using the improved explosive welding technique, where the thin Al plate is successfully bonded onto the thin Mg plate, and the surface of the Mg plate appears to have the same appearance of the as-received condition without any wrinkles or large deformations. The reason for this good weld is due to the combined action of the improved explosive welding technique and careful control of the welding parameters, which effectively protect the composite plate from the damage of shock waves as well as impede its deformation in the welding process. In addition, a small number of cracks were found at the end of the clad, which are perpendicular to the detonation direction, as shown in Figure 5a. This result is consistent with previous work, where the macromorphology of Mg/Al explosive welding plates was also studied [22]. These cracks might be caused by tensile waves. Because of the low strength of the Mg plate and its hydrodynamic behavior in the collision process, the reflected tensile waves can easily tear the thin Mg plate. According to Figure 5b (the red zone represents a well bonded joint, while blue means the opposite), the poor bonding zones are mainly concentrated at the detonator zone, and occasionally situated at the end. It should be noted that the poor bonding in these two regions is a common issue, which can be found in almost all explosive welding systems.

The reason for this is due to the unstable detonation in the detonator region and the rarefaction wave reflected at the boundary of the welded plates.

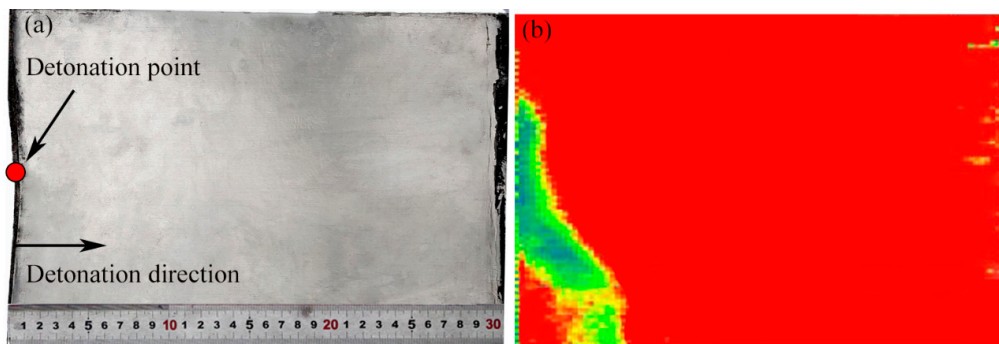

**Figure 5.** Macroscopic feature of AA5052/AZ31B plate, (**a**) photograph of AZ31B surface, (**b**) ultrasonic testing results.

## 4.2. Microstructure of Bonding Interfaces

Figure 6 shows the microstructure of the AA5052/AZ31B interface obtained using SEM, which confirmed that a high-quality metallurgical bond was created without voids and cracks at the bonding interface. According to Figure 6a, the AA5052/AZ31B interface shows a regular wavy structure, in which the amplitude and period can be determined to be ~160 μm and ~40 μm, respectively. The formation of the wave can be attributed to the compression waves generated by the collision at the interface, which propagates periodically in the form of S-waves and P-waves near the interface that overlap with each other, consequently forming a periodic disturbance along the interface and eventually promoting the formation of the wavy interface [35]. Compared to the straight interface that can be also obtained in explosive welding, the wavy interface is usually preferred due to better mechanical properties and more weld area [36,37]. Figure 6b shows single wave structures using high magnifications, where a typical melted zone was found within the wave vortex area (marked by the white line). During the high-speed oblique collision process, a jet flow with a high temperature was formed at the interface due to adiabatic shear action of the interface metals. Because of the rapid closing process between the base and flyer plates, part of the jets were trapped by the parent metals, leading to formation of these melted zones. Owing to the high temperature and intense mixing of multiple metal components, the melted zone often shows a complex microstructure such as formation of amorphous, nanocrystalline and intermetallic compounds and defects, which affect the bonding properties in a negative manner [38].

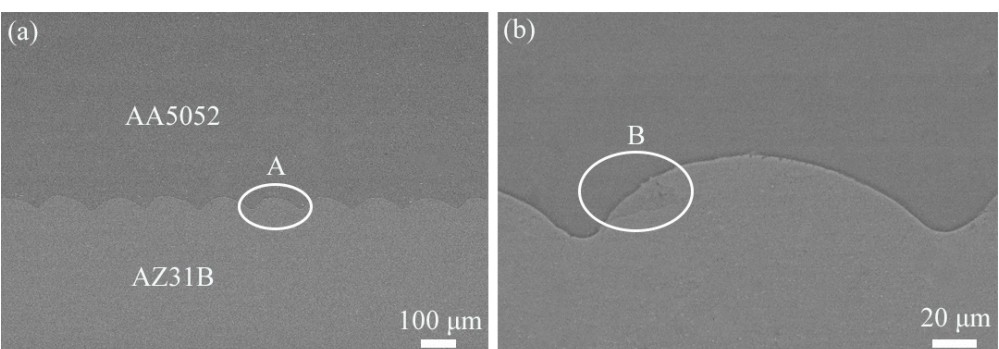

**Figure 6.** SEM image of AA5052/AZ31B bonding interface, (**a**) typical wave, (**b**) high-resolution image of position A in (**a**).

### 4.3. Tensile Tests

The tests were repeated three times; one of the typical stress-strain curves of the Mg/Al composite plate along with the fractured specimen is shown in Figure 7, from which the tensile strength and elongation can be determined. The tensile strength of the three specimens attained was 341 MPa, 326 MPa and 353 MPa, whose uncertainty was calculated to be 340 ± 7.8 MPa. Elongation of the specimens was measured to be 23%, 20.5% and 24.3% with uncertainty of 22.6% ± 1.1%. The uncertainty results indicate the good repeatability of the test. In addition, the tensile strength of the composite plate is higher than that of the 5052Al (300 MPa) and ZA31B (240 MPa), which is caused by the work hardening during the welding process. Due to the strength and toughness of the Mg and Al plate, they differ greatly, and the bonding quality is poor in some previous works [7,21,39]; a span can be observed during the stress drop stage in the stress-strain curves, and interfacial debonding can be found after the tensile test. However, this phenomenon has not been observed in this work. It can also be seen from the fracture surface in Figure 7 that the two welded materials fracture simultaneously, and no separation occurs at the bonding interface, indicating that a good adhesive strength was obtained between the Mg and Al plate.

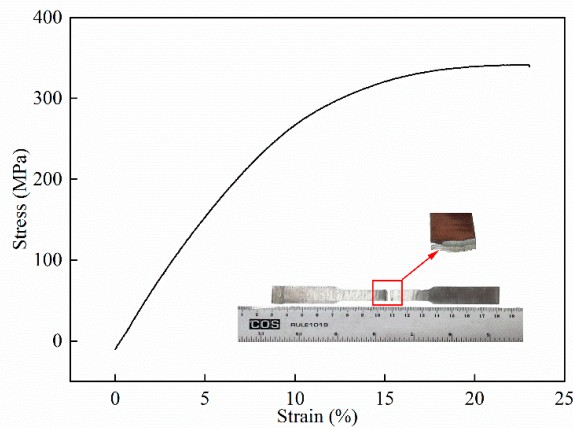

**Figure 7.** Typical stress-strain curves obtained by tensile tests.

### 4.4. Tensile Shear Tests

Shear strength is an important index for assessing the welding interface and general bonding quality [40]. Figure 8 shows the typical stress-distance curves and specimen obtained using tensile-shear tests, where fracture took place in the AZ31B side, indicating that the tensile-shear strength value of the AZ31B/5052Al interface is beyond 85 MPa, which is larger than the value of 60–70 MPa obtained in previous works [41,42]. The reason for the high bond strength in this work could be due to excellent bonding quality and the higher levels of hardening in the thin plate explosive welding.

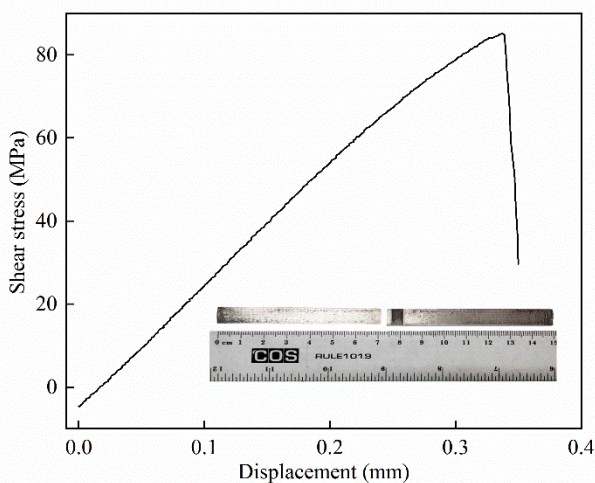

**Figure 8.** Typical stress-distance curves obtained by tensile-shear tests.

## 5. Simulation of Welding Process with SPH Method

The SPH method was applied in this part to help to get a better understanding of the accuracy of the approach that determines the welding parameters through numerical simulation. SPH is capable of treating problems with large material distortions due to its meshfree and particle-based characteristics [43]. What is more, typical features like a wavy bonding interface, jetting phenomenon and vortices in the explosive welding process can be reproduced, which are pretty effective for the validation.

In this case, the 2D model was selected for calculation, and the initial conditions were simplified in the form of the initial oblique angle and velocity of the flyer plate that were derived from Table 1 ($V_p$ equals 620 m·s$^{-1}$, $\beta$ is 12.7°) to save computational costs. The particle size is extremely important in visualizing the interface and jetting profiles, as well as maintaining the stability of computation, which is set as 5 μm here. To save computation costs, the size of both of the flyer plate and the base plate are 0.5 mm in thickness and 10 mm in length, which means that over 400,000 particles were used for modelling.

Figure 9 well represents the explosive welding process; a wavy joint interface is clearly observed with a wave length of 150~175 mm and wave amplitude of 28~35 mm, which are pretty close to the values obtained by the experiment, indicating that the numerical simulation is well congruous with the experiment. The phenomenon of jet can also been seen clearly in the SPH simulation, which is an essential feature for the formation of the wavy bonding. In addition, the diffusion of elements across the joint interface was detected, which is generally regarded as a good sign for metallurgical welding with excellent mechanical properties.

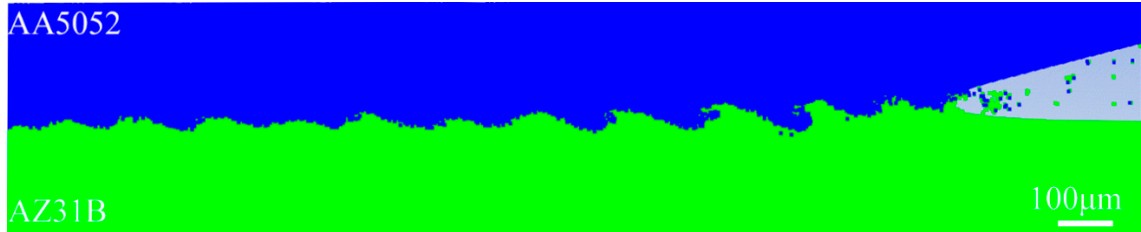

**Figure 9.** Interfacial morphology obtained using the SPH method.

The temperature distribution of the AA5052 and AZ31B is displayed in Figure 10a. The result reveals that the high temperature zone is concentrated near the bonding interface, which is caused by the adiabatic compression of air between the plates. The temperature in some regions (as indicated

by the white boxes) has reached 1500~1800 K, which exceeds the melting temperature of both of the materials, leading to the occurrence of the molten zone. This is illustrated by the experimental results, as indicated in Figure 6b. Jetting is produced with temperatures beyond 2000 K. Figure 10b shows a localized wavy band of plastic strain formed at the joint region; the highest strain can reach up to 3.3 around the interface, which is consistent with the results obtained by Bondar et al. [44]. The pressure contour is depicted in Figure 10c, which must be larger than the dynamic elastic limit of the plates to ensure the formation of jetting in the interface. The C-J pressure obtained by Autodyn is about 2.2 GPa, as shown in Figure 10c, closing to the theoretical value of 2.08 GPa, indicating a good compatibility between the numerical simulation and theoretical calculation. As shown in Figure 10d, the magnitude of the shear stress predicted for the bonded area in the SPH simulation is much higher than that for the non-bonded area. The value of shear stress in the flyer plate is greater than that of the base plate; similar results have been achieved in previous explosive welding simulations with the SPH method [45].

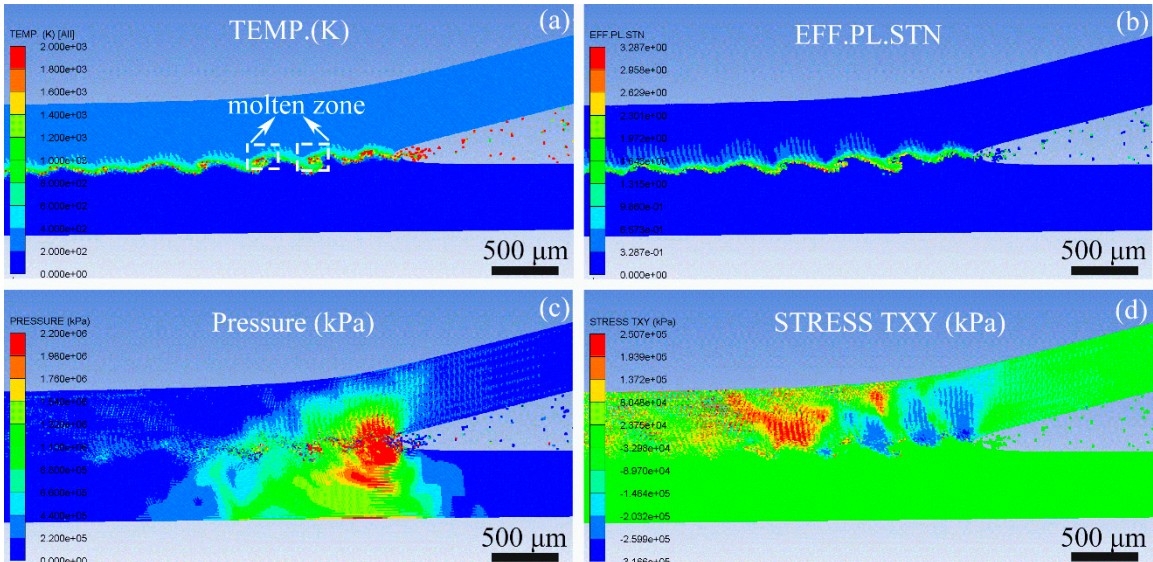

**Figure 10.** Interface between AA5052 and AZ31B, (**a**) temperature in Kelvin using SPH, (**b**) plastic strain using SPH, (**c**) pressure using SPH, (**d**) shear stress using SPH.

## 6. Conclusions

In this work, a thin Mg/Al composite plate with a total thickness of 1 mm was successfully prepared using an improved explosive welding technique and optimum parameters obtained by theoretical analysis and numerical simulation. The bonding properties were investigated using microstructure observation and mechanical testing, and the interfacial behavior was simulated using the SPH method. The following conclusions can be drawn:

(1) The improved explosive welding technique is an effective method for producing a thin Mg/Al composite plate. After the explosion, the composite plate showed a high surface quality, and the poor bonding zones are only situated at the detonator zone and the end.

(2) The microstructure analysis shows an excellent metallurgical bond at the AA5052/AZ31B interface, where a regular wavy structure was formed without voids and cracks.

(3) The AA5052/AZ31B interface shows good mechanical properties, in which no separation was found between the Mg and Al plate, and the interfacial shear strength reached 85 MPa.

(4) SPH simulation revealed an extreme condition of high pressure, local high temperature and large deformation in the welding process, and the characteristic parameters of waves obtained by simulation are well congruous with the experiment.

**Author Contributions:** Designed project, X.L.; analyzed the data and wrote the paper, Q.W.; revised the paper, B.S.; performed the explosive welding experiment, Q.W. and Y.W. All authors have read and agreed to the published version of the manuscript.

**Funding:** This research was supported by the Natural Science Foundation of Anhui Province of China (grant No. 1808085QA06), the Scientific Research Foundation of the Education Department of Anhui Province of China (grant No. KJ2018A0090) and the Postdoctoral fund of Anhui province of China (grant No. 2019B355).

**Conflicts of Interest:** The authors declare no conflict of interest.

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
