# Peer review of "Experimental and Numerical Studies on Preparation of Thin AZ31B/AA5052 Composite Plates Using Improved Explosive Welding Technique"

_metals, doi:10.3390/met10081023_

Round 1
Reviewer 1 Report
The manuscript entitled ‘Experimental and numerical studies on preparation of thin AZ31B/AA5052 composite plates by improved explosive welding technique’ falls within the scope of the journal Metals. The paper contains very interesting numerical and experimental results. It is of sufficient scientific interest and has originality in its technical content to merit publication. The authors have cited the relevant literature. Methods, interpretations of results and conclusions are correct and novel. The issues were well presented. In terms of content, the analysis does not raise any objections. The arrangement of work maintains substantive continuity and constitutes a logical whole. However, the manuscript is not suitable for publication in its present form and requires small editorial corrections.
Comments and remarks are presented below.
In the title, some words start with a capital letter and some begin with a lowercase letter.
SPH abbreviation should be explained in detail on the first use.
Authors should carefully check and correct References, especially positions: 4, 5, 20-23, 39, 42.
Reviewer 2 Report
The article entitled "Experimental and Numerical Studies on Preparation of Thin AZ31B/AA5052 composite plates by improved explosive welding technique " written by Qi Wang et al deals with explosive welding of Al alloy and Mg alloy to obtain a composite plate offering a new structural element, lighter with higher mechanical and physico chemical performances than single Al plates. The presented technique brings a solution for a low cost production of Al/Mg composite plates. There is thus interests for the scientific and industrial community. In addition, the matter tackled in this article also present some metallurgical aspects on the bonding interface and thus is fully in the scope of "Metals" journal.
Based on the above comments, I am in favour of publishing this article in metals magazine, after some major corrections:
- Introduction :
Please announce the structure of the paper at the end of the introduction
- section 2.1:
p3, line 100, please specify the density of the ANFO utilized for your experiments, and also its prills size.
p3, line 100, please explain how is the velocity detonation determined.
p3, line 101, what do you mean by "destruction of detonation" ? is it "spalling" ?
p3, line 109, please give the reference where you took the JWL paramteres for ANFO
p3 line 111, the Mie-Grüneisen equation is the first equation (3). The shock Hugoniot EoS is the relation Us=C0+sUp that is missing in your text, you shall add it and also please specify the refs where you took the parameters C0 and s. Also specify that Gamma0*rho0=gamma*rho, it is a convennient assumption always made in this kind of approach since there is no trivial relation for Gamma.
p3 line 116, add subscript 0 for gamma.
p3 line 117-118, add refs for Al and Mg coefficients, they look exotic, usually, s=1.339 for Al, you mention 1.4, it is quite different. Also the bulk sound velocity is usually 5386 m/s for Al, you mention 6305 m/s, what is closer to the longitudinal elastic velocity, please clarify.
p4, line125-127, why did you chose these thicknesses ? could you please explain ?
p4, line 132, please "very" in the sentence "...instantly to a very high speed of ..."
p4, line 136, typographic mistake "...with the increase of ..."
p4 fig 2, you show velocities that are not stabilized 'still increasing with time', thus it should be introduced a notion of discrepencies in the terminal velocity. Please show the explosive thickness in the legend, instead of group 1-4.
p4 line 139, the numbering of the header is also 2.1, it should be 2.2 Determination of welding parameters.
p5, fig 3, justify the interest of studing the left border of the diagram.
Please specify the size elements, constitutive laws for Al and Mg, interface definition, symmetry conditions if any, ...
- section 3.1
Please specify whether the tensile tests are performed in force controle or displacement controle. What is the metrology utilized for measuring displacement ? extensometer ? Digital Image correlation ?
You shall also mention the Roughness of the plates before welding
- section 4
p6, line 183, "Mg plate and its fluid like behavior..." it should be "Mg plate and its hydrodynamic behavior..."
p7, line 205, remove "ultra" in "the jet flow with ultra high temperature ..."
p7, line 210 and fig 6, inidcates the intermetallics that are likely to be detected. Did you perofrmed SEM analysis with probe for identification ?
p 7 line 215 it should be section 4.3
p 7 line 217 Please give uncertainties for the tensile strength (+ or - something) as well as for the elongation. Also mention the repetability of the tests.
p8 line 226, it should be 4.4 tensile shear tests
- section 5
p8, line 238, typographic mistake "regarded as a convenient"
p8 line 239, typo mistake "...and meshfree characteristics make it possible..."
In your simulation, why did not you modelled the anvil nor the buffer ? could you please justify your choice and discuss it ?
fig 9 and 10, please add scale
I would appreciate a comparison of the pressure induced at impact with the shock polar technique.
Also the theoretical Chapman-Jouguet pressure resulting from the detonation should be discussed with the one computed by autodyn.
- conclusion
line 286, typo mistake "...revealed an extreme condition" and remove "ultra"
Reviewer 3 Report
The paper tried to weld relatively thin Mg alloy and Al alloy using explosive welding technique. The followings are considered weak points of the paper at this stage.
- Line 62; “Daisuke et al.” seems “Inao et al.”
- Line 75-76; “….Mg plate is rigidly constrained by a steel plate….” It is not match with Fig.1. If this is authors’ one of the improvements, it is recommended to be emphasized in the paper.
- Equation (4) should be appeared earlier.
- Line 125; Authors are used thin explosive layer ranging from 6 to 7.5 mm. Thinking about the critical diameter of ANFO explosive, the explosive is considered not fully detonated. The numerical simulation is not acceptable by the reviewer’s impression.
- Line 130; flyer plate thickness is 1mm including buffer Al. The thickness of AZ31B is not written, though it seems 0.5mm.
- Fig. 8 and the related part in the manuscript; It is not clear but it seems that the tested sample was fractured in aluminum by tensile load. Nothing was mentioned in the manuscript and should be discussed comparing the strength of both phases.
- Fig.10 (a); Temperature is not possible to read by the figure. How could the authors decided the two areas marked "molten zone"?
- The paper includes just one experimental result. More experiments recommended.
Round 2
Reviewer 2 Report
dear authors, thank you for your valuable improvements.
I am still in conflict with the value of 6305 m/s for Al. You call it Bulk velocity, line 125 but in response number 8, you also call it longitudinal velocity, what is not the same value. After having read the refs 29 and 30, Bulk velocity is rather around 5350 m/s and longitudinal velocity more like 6300 m/s so it should be modified.
Author Response
Point 1: I am still in conflict with the value of 6305 m/s for Al. You call it Bulk velocity, line 125 but in response number 8, you also call it longitudinal velocity, what is not the same value. After having read the refs 29 and 30, Bulk velocity is rather around 5350 m/s and longitudinal velocity more like 6300 m/s so it should be modified.
Response 1: Thank you for your valuable comment. We confused the concept of bulk velocity and longitudinal velocity, the mistake has been amended in your help. The bulk velocity (c0) has been modified to 5368 m/s according to many previous researches that is the same as your advice, the influence of the change has also been added, though it is not obvious, i.e. (velocity of flyer plate calculated in Fig.2, weldability window in Fig.3).
Thank you sincerely!